# A study protocol for comparing the treatment of varicose tributaries either concomitantly with or separately from endovenous laser ablation of the incompetent saphenous trunk (the FinnTrunk Study). A multicenter parallel-group randomized controlled study

Jaakko Viljamaa[1,2], Khalil Firoozi[1,2], Maarit Venermo[3], Matti Pokela[4,5], Toni Pihlaja[4,5], Karoliina Halmesmäki [3], Harri Hakovirta [1,2,6]*

1 University of Turku, Turku, Finland, 2 Department of Vascular Surgery, Turku University Hospital, Turku, Finland, 3 Department of Vascular Surgery, University of Helsinki and Helsinki University Hospital, Helsinki, Finland, 4 Department of Vascular Surgery, Oulu University Hospital, Oulu, Finland, 5 University of Oulu, Oulu, Finland, 6 Satasairaala, Pori, Finland

* haheha@utu.fi

## Abstract

### Background

Opinions on the treatment of varicose tributaries in relation to saphenous ablation in varicose disease vary. Moreover, the possible role of the tributaries regarding the recurrence of varicose disease remains unclear. The aim of the FinnTrunk study is to compare two different treatment strategies for varicose disease in a randomized setting. In group one, the initial treatment will entail endovenous laser ablation (EVLA) of the incompetent saphenous trunk without tributary treatment. In group two, the varicose tributaries will be treated with ultrasound-guided foam sclerotherapy (UGFS) concomitantly with truncal ablation. The primary outcome measure is the need for additional procedures during the follow-up. The secondary outcome measures are the cost of treatment and recurrence of varicose disease.

### Methods

Consecutive patients with symptomatic varicose disease (CEAP clinical class $C_2$–$C_3$) will be screened for the study. Patients who fulfil the study criteria and give their informed consent will be scheduled for the procedure and randomized to either study group. Patients will be followed-up at 3 months, 1 year, 3 years, and 5 years. The post-procedure pain score based on a numeric rating scale (NRS) and also the use of analgesics, as well as possible procedure-related complications will be recorded at 3 months. Patient-reported outcome measures (PROMs) will be recorded at 1 year. Data pertaining to the additional treatment of varicose tributaries, the Aberdeen Varicose Vein Questionnaire (AVVQ), the Venous Clinical Severity Score (VCSS), and the health-related quality of life (EQ-5D-5L) will be collected

**Data Availability Statement:** No datasets were generated or analysed during the current study. All relevant data from this study will be made available upon study completion.

**Funding:** The funders did not and will not have a role in study design, data collection and analysis, decision to publish, or preparation of the manuscript. H.H. Varsinais-Suomen Rahasto, Award number: 85211811 H.H. Satasairaala, Pori (federal fund), Award number: (None).

**Competing interests:** he authors have declared that no competing interests exist.

at each follow-up visit. A duplex ultrasound (DUS) examination will be performed at each visit, and data on varicose tributaries and the need for additional treatment will be recorded.

## Trial registration

Registered on ClinicalTrials.gov, ID NCT04774939.

## Introduction

Varicose veins are common [1], and they can be responsible for various symptoms [2], which can be bothersome to the patient and cause them to seek treatment. The most frequent hemodynamic abnormality in primary varicose disease is reflux of the great saphenous vein (GSV), the anterior accessory saphenous vein (AASV), or the small saphenous vein (SSV) [3–5]. When present, saphenous reflux is considered the primary target of treatment [6], with endovenous thermal ablation and cyanoacrylate closure being the preferred treatment modalities [7–11]. Varicose tributaries connected to the incompetent saphenous trunk can be treated with phlebectomy or ultrasound-guided foam sclerotherapy (UGFS), either concurrently with the truncal ablation (combined approach), or as a separate procedure (staged approach) [12].

Both of the abovementioned approaches have their advantages [12], and both have been associated with good outcomes and low complication rates in the clinical setting, and also in the research setting [13]. However, there is a paucity of randomized controlled trials that have addressed the treatment of varicose tributaries in relation to truncal ablation [14–16], and much of the evidence for the combined approach recommended by the European and American clinical practice guidelines [10, 11] comes from registry data, which may be incomplete and susceptible to bias [6, 17]. Furthermore, not all patients would need, or for that matter want, their varicose tributaries to be treated, if their symptoms could be or had already been adequately eliminated by truncal ablation alone [10, 11, 15, 16, 18, 19].

The aim of the FinnTrunk study is to compare the outcomes of two different treatment strategies for varicose disease. Patients will be randomized to either endovenous laser ablation (EVLA) of the incompetent saphenous trunk alone, or EVLA of the saphenous trunk in combination with UGFS of the varicose tributaries. Patients will be followed for up to five years. The primary outcome measure will be the need for additional treatment of the varicose tributaries during the follow-up, as judged by the patient. The secondary outcome measures will be the cost of treatment and recurrence of varicose disease.

## Methods

### Study design and data management

The FinnTrunk study is a non-blinded, multicenter, parallel-group, randomized, controlled study, which will be conducted in four large Finnish hospitals (Helsinki University Hospital, Oulu University Hospital, Satasairaala hospital in Pori, and Turku University Hospital). The participating hospitals were selected based on the high volume of endovenous procedures in their respective vascular surgery units. Data will be collected by study investigators at each participating center and uploaded to an electronic data capture system (REDCap, Vanderbilt University, Nashville, TN, USA) hosted by and at Helsinki University Hospital. The FinnTrunk study was registered on ClinicalTrials.gov on 24 February 2021 (study identifier

**Table 1. Study questions.**

**The main study question**
  • Is thermal ablation of an incompetent saphenous trunk adequate as a stand-alone treatment in varicose disease, or do the varicose tributaries also need to be treated in the same procedure?
**The questions used to approach the main question**
  • Does treating varicose tributaries concomitantly with saphenous ablation have an impact on the short-term results of the treatment?
  • What proportion of the patients whose varicose tributaries that have or have not been treated concomitantly will need further treatment of their varicose disease?
  • Is the timing of the tributary treatment important in relation to the quality of life of the patient?
  • Is the tributary treatment or its timing important in relation to the longer-term outcomes or recurrence of varicose disease?
  • Does the selected treatment strategy have an effect on the treatment costs?

NCT04774939), and the first patient was entered in the study register on 6 October 2021. The first study results will be published in a peer-reviewed journal in 2024.

## Aims of the study

The aim of the FinnTrunk study is to compare two different treatment strategies for varicose disease. More specifically, the aim is to evaluate the possible benefits of the combined approach (EVLA of the incompetent saphenous trunk with concomitant treatment of its varicose tributaries by UGFS) and compare it to the staged approach (EVLA of the saphenous trunk as a stand-alone therapy and then UGFS of the varicose tributaries at follow-up visits as necessary). The study questions are listed in Table 1.

## Study setting and patient screening

Consecutive patients referred to the participating vascular surgery units with symptomatic varicose disease in one leg will be screened for the study by clinical assessment and duplex ultrasonography (DUS). The study inclusion and exclusion criteria are listed in Table 2.

**Table 2. Inclusion and exclusion criteria.**

**Inclusion criteria**
  • Age of patient 18–80 years.
  • Symptomatic varicose disease in one leg (designated as the study leg).
  • CEAP clinical class $C_2\neg$–$C_3$ (or $C_{4c}$), with the provision of recruiting patients in the clinical classes $C_{4a}$–$C_{4b}$ later in the study, if necessary (slow recruitment); CEAP etiologic, anatomic, and pathophysiologic class $E_p$ $A_{s,d}$ $P_r$ [20].
  • Incompetent great saphenous vein (GSV) and/or anterior accessory saphenous vein (AASV) or small saphenous vein (SSV) in the study leg, which is suitable for treatment with EVLA.
  • Varicose tributaries connected to the incompetent saphenous trunk.
  • No prior venous procedures on the study leg.
  • Palpable posterior tibial artery (PT) or dorsalis pedis artery (DP) pulse / If not: ankle-brachial index (ABI) >0.75.
**Exclusion criteria**
  • Body mass index (BMI) ≥35.
  • Moderately or severely symptomatic (NYHA class III–IV) congestive heart failure (CHF).
  • Patent foramen ovale (PFO).
  • Significant lymphedema of the study leg.
  • Prior deep vein thrombosis (DVT) or thrombosis of the muscular veins in the calf of the study leg, or pulmonary embolism (PE).
  • Extensive deep vein reflux in the study leg.
  • Venous disease of the contralateral leg, for which treatment is indicated.
  • Pregnancy or lactation, or patient planning pregnancy within the following 6 months.
  • Known thrombophilia.
  • Known allergy to lidocaine or a sclerosant.

**Clinical assessment.** 'Symptomatic' designates symptoms that can be attributed to varicose disease. Examples of such symptoms include pain or aching (or a comparable symptom, e.g., heaviness or tiredness) and edema of the leg, which typically manifest over the course of the day while the patient is prone; cramps and restless legs (these symptoms can also manifest at rest during night-time); in addition to tingling, numbness, and itching. The inclusion criteria do not define which of these symptoms the patient should present with or how disturbing the symptoms should subjectively be for the treatment to be considered indicated. The decision to treat will be left to the discretion of the clinician in charge of the patient, who will assess whether the entirety of the clinical picture (symptoms, clinical signs, and DUS findings) is such that the patient, according to the clinician's experience, would benefit from the treatment.

The study will primarily recruit patients with unilateral symptomatic varicose disease, and the symptomatic leg will be designated as the study leg. However, patients with bilateral disease where the contralateral leg also has varicose veins or lesser findings of chronic venous disease (telangiectasias or reticular veins) may also be recruited, if treatment of the contralateral leg is not deemed to be indicated by either the patient (based on the symptoms) or by the clinician (based on other indications for treatment, such as skin changes). The presence of the aforementioned symptoms in the contralateral leg will be allowed, provided that the pain attributable to venous disease or the symptoms comparable to pain (*other discomfort* in the Venous Clinical Severity Score [VCSS] [21]) are only occasional (VCSS 1 mild pain) and any edema of presumed venous origin is limited to the foot and ankle (VCSS 1 mild edema). Moreover, varicose veins in the contralateral leg must be few and scattered (VCSS 1 mild varicose veins). If there are multiple varicosities in the contralateral leg, they must be confined to the calf *or* the thigh (VCSS 2 moderate varicose veins).

A patient will not be recruited into the study, if in their contralateral leg they have varicose disease requiring treatment, as deemed necessary by the patient or the clinician in charge of the patient. Furthermore, a patient will not be recruited into the study, if the pain or symptoms comparable to pain in the contralateral leg occur daily (VCSS $\geq 2$), the edema extends above the ankle (VCSS $\geq 2$), or there are multiple varicosities that involve *both* the calf and the thigh (VCSS 3). DUS of the contralateral leg will not be mandatory.

**Duplex ultrasonography (DUS).** The protocol for the DUS examination follows previously published recommendations [22, 23]. During the examination, the patient will remain in the standing position with their weight on the opposite leg and with the leg under investigation relaxed. Reflux in the vein will be preferentially elicited by a pneumatic cuff, but manual compression is also allowed. The cut-off value for reflux will be 0.5 seconds in the saphenous veins and 1 second in the deep veins [10, 11, 24]. The US probe will be held longitudinally to the vein being investigated, as the longitudinal view is more accurate for assessing reflux than the transverse view [22]. The following veins and junctions will be assessed for reflux: the common femoral vein (CFV), the popliteal vein (PV), the saphenofemoral junction (SFJ), the saphenopopliteal junction (SPJ), the great saphenous vein (GSV) in the thigh, the anterior accessory saphenous vein (AASV), the small saphenous vein (SSV); and any other veins as indicated.

As stated in the inclusion criteria, the patient must have reflux in the GSV and/or AASV, or in the SSV, to be eligible for the study. Additionally, the reflux must originate at the usual anatomical junction of that vein with the deep vein, i.e., the SFJ for the GSV and the SPJ for the SSV. If the most proximal refluxing saphenous vein in the thigh is the AASV that joins the GSV at or distally to the SFJ, then the reflux may originate at the SFJ or at the junction of the AASV with the GSV. If the AASV has a separate junction with the femoral vein, the reflux must originate at that junction.

The diameter of the GSV will be measured at three points: 3 cm distal from the SFJ, at the mid-thigh level, and at the level of the knee joint. The SSV will be measured at two points: 3 cm distal from the SPJ and at the mid-calf level. In keeping with the frequently shorter course of the AASV, it will be measured at one point, regarded as representative of the vein diameter. The US probe can be held either longitudinally or transversally during these measurements. All three saphenous trunks will be assessed and the diameter, patency, and possible reflux of both the GSV and SSV will be recorded for all study patients. The corresponding parameters for the AASV will be registered only when it is the target for ablation.

Perforating veins will not be systematically searched for or recorded, as there is no agreement regarding their significance in uncomplicated venous disease and their treatment in this setting is rarely considered to be indicated [25, 26].

Extensive deep vein reflux will be defined as reflux of *both* the CFV and PV.

## Patient enrolment and randomization

The patients who fulfill the study criteria will be given a brief information letter and offered the possibility to participate in the study. The baseline characteristics of the patients who give their written informed consent will be recorded in the study register. The patients will fill in the Aberdeen Varicose Vein Questionnaire (AVVQ) [27] (with question 1 omitted [28, 29]) and the EQ-5D-5L, while the investigator will ascertain and record the patient's Venous Clinical Severity Score (VCSS) [21], Venous Disability Score (VDS) [30], and the clinical class of the Clinical-Etiology-Anatomy-Pathophysiology (CEAP) classification [20], in addition to recording the location and extent of the varicose tributaries.

The patients will be randomly allocated with the REDCap to one of the two study groups before the procedure. Patients in group one will be treated with EVLA of the incompetent saphenous trunk without treatment of the varicose tributaries. Patients in group two will be treated with UGFS of the varicose tributaries concomitantly with the EVLA.

If the patient wishes to decline participation in the study, they will be treated according to the usual regime of the unit.

## Interventions

A diode laser (Leonardo® Dual 45 or Ceralas® E) with a wavelength of 1,470 nm and equipped with a radial fiber (ELVeS® Radial® 2ring Fiber, ELVeS® Radial® 2ring slim Fiber, or ELVeS® Radial® Fiber) (biolitec® AG, Vienna, Austria) will be used for the EVLA. The procedure will be performed under DUS guidance and according to the manufacturer's instructions, and the goal will be to treat the longest segment of the incompetent saphenous trunk as feasible, taking into account the anatomical relationships of the vein with the nearby sensory nerves and other aspects of patient safety. Accordingly, the GSV will be cannulated for EVLA in the thigh or in the proximal third of the calf, the SSV will be cannulated in the proximal half of the calf, and the AASV will be cannulated in the thigh. The length of the treated vein will also be recorded.

In brief, the target vein will be cannulated percutaneously, and the laser fiber will be advanced in the vein to approximately 1 cm distal from the junction with the deep vein. Perivenous tumescent anesthesia (NaCl 500 ml + lidocaine 1% cum adrenalin 50 ml + sodium-bicarbonate 7.5% 10 ml) will be infiltrated around the vein, and the position of the fiber tip will be verified. The patient will be positioned in the Trendelenburg position during the withdrawal of the fiber, and the speed of withdrawal will be adjusted to reach a linear endovenous energy density (LEED) of approximately 70 J/cm. The patients allocated to group 2 (concomitant UGFS) will be treated with 1% polidocanol (Aethoxysklerol®, Kreussler Pharma,

Wiesbaden, Germany) foam (1 ml of sclerosant mixed with 3 ml or air) after the laser ablation. The volume of foam will be limited to 20 ml. The patency of the deep veins will be verified by using US after the procedure.

The puncture sites will be covered with adhesive pads, which the patient can remove on the following day. A thigh-length class II compression stocking will be applied, and the patient will be instructed to wear the stocking for 1 week post-procedure (round the clock for the first 2 days and during daytime for the following 5 days thereafter).

Details of the procedure and possible immediate complications will be recorded in the study register.

Varicose tributaries can be treated with UGFS as described previously at any follow-up visit as necessary. A thigh-length class II compression stocking will be worn for 1 week post-procedure.

## Outcome measures and follow-up

The duration of the follow-up will be five years. The primary outcome measure will be the need for additional treatment of the varicose tributaries during the follow-up, as judged by the patient. The secondary outcome measures will be the cost of treatment and recurrence of varicose disease. Other outcome measures of interest will the length of the procedure time and the impact of treatment on the Aberdeen Varicose Vein Questionnaire (AVVQ) score, the Venous Clinical Severity Score (VCSS), health-related quality of life (the EuroQoL EQ-5D-5L) score, and patient-related outcome measures (PROMs). Data pertaining to PROMs will be collected at 1 year with two questions: "Was the treatment you were given of good quality?" and "Did the treatment you were given meet your expectations?"

In addition, the following data will be collected at the 3-month follow-up visit: pain score based on numeric rating scale (NRS) during the procedure and at three time points post-procedure, the use of analgesics, and the possible procedure-related complications. Patients will also be asked about the duration of sick leave and for their estimate of the appropriate recovery time.

The study leg will be assessed using DUS at each follow-up visit. The protocol for the examination will be identical with the diagnostic DUS described previously under Study setting and patient screening, i.e., the diameter of GSV and SSV will be measured and both veins will be assessed for patency and reflux regardless of whether the vein has been treated or not; the AASV will be systematically assessed only when it has been the target for ablation. Apart from that, it will be assessed as needed, i.e., if there are varicose tributaries that could be connected to the AASV. The treated saphenous trunk will be assessed according to the recommendations and definitions of De Maeseneer et al. [23]. Data on the saphenous trunks and varicose tributaries will be recorded in the study register. During the follow-up visits, patients in both study groups will be able to receive UGFS, if judged necessary by the patient, for the remaining or new tributaries. The additional treatments will be recorded in the study register.

A schematic diagram of the timeline of the study [31] is presented in Fig 1.

## Statistical analysis

The sample size calculation for the study was based on the systematic review and meta-analysis by Aherne et al. [13]. Combined treatment of the saphenous trunk and varicose tributaries results in a 6.3% and staged treatment in a 36.1% reintervention rate. According to the power analysis, statistical dichotomous analyses between two independent groups with an alpha of 0.01, and a power of 80%, thus a 1:1 randomization would result in a sample size of 84. When calculated for a 12% difference, the anticipated sample size would, accordingly, be 228.

| | STUDY PERIOD | | | | | |
|---|---|---|---|---|---|---|
| | ENROLMENT | ALLOCATION | FOLLOW-UP | | | |
| TIME POINT | | 0 | 3 months | 12 months | 36 months | 60 months |
| Eligibility screen | x | | | | | |
| Informed consent | x | | | | | |
| Allocation | | X | | | | |
| **INTERVENTIONS** | | | | | | |
| EVLA | | $x^1$ | | | | |
| EVLA + UGFS | | $x^2$ | | | | |
| UGFS | | | x* | x* | x* | x* |
| **ASSESSMENTS** | | | | | | |
| Baseline characteristics | x | | | | | |
| DUS | x | | X | x | x | x |
| CEAP | x | | | x | x | x |
| VCSS | x | | X | x | x | x |
| VDS | x | | | x | x | x |
| EQ-5D-5L | x | | x | x | x | x |
| AVVQ | x | | x | x | x | x |
| PROMs | | | | x | | |
| Pain post-procedure | | | x | | | |
| Complications | | X | x | | | |
| Varicose tributaries | x | | | x | x | x |
| Additional treatment | | | x | x | x | x |

**Fig 1. Schedule of enrolment, interventions, and follow-up of the FinnTrunk study.** The specific time points are listed in the columns and the events are given in the rows. Participants will be randomly allocated to EVLA without ($x^1$) or with ($x^2$) concomitant UGFS before procedure at the treatment visit. All participants will have a follow-up visit at 3 months, 1 year, 3 years, and 5 years. If found necessary, the varicose tributaries can be treated at any of the follow-up visits. Abbreviations: the Aberdeen Varicose Vein Questionnaire (AVVQ), (the clinical class of) the Clinical-Etiology-Anatomy-Pathophysiology classification (CEAP), endovenous laser ablation (EVLA), Duplex ultrasound (DUS), EuroQoL quality of life questionnaire (EQ-5D-5L), patient related outcome measures (PROMs), ultrasound-guided foam sclerotherapy (UGFS), the Venous Clinical Severity Score (VCSS), the Venous Disability Score (VDS). [1]According to the randomized allocation (group 1) [2]According to the randomized allocation (group 2) *If judged necessary by the patient (both groups).

Considering the long follow-up (5 years) and the anticipated large number of dropouts, the sample size was set at 260 (130 for each group).

For dropouts, the results will be analyzed until their last follow-up visit. If a patient has not completed their treatment or has not been treated according to their allocation (crossover), their participation in the study will be terminated and their data will not be included in the analysis.

All statistical analyses will be performed using SPSS version 27 for Mac (IBM, Armonk, NY, USA). Descriptive statistics will be used to summarize patient baseline characteristics. Mean values and standard deviations (SD) will be applied for continuous variables if normally distributed. Correspondingly, median values and interquartile range (IQR) will be applied for non-parametric variables. The Fisher's exact test will be used to compare categorical variables, and the Wilcoxon Signed-Rank Test and the Mann-Whitney U test will be used to compare ordinal variables. Normally distributed continuous variables will be compared by using the Student's t-test for values previously tested using the Shapiro–Wilk test. A $p$-value of $< .05$ will be considered statistically significant.

## Discussion

Despite the clinical practice guidelines' preference for a combined approach [10, 11], the opinions about the treatment of varicose tributaries in relation to saphenous ablation vary, and these differences may not necessarily be attributable solely to differing interpretations of the available evidence [32]. Neither should these varying opinions come as a surprise, considering the generally benign course of varicose disease and the goals of its treatment [33, 34] in addition to patients' preferences and expectations regarding their treatment [35]. That said, we hope that the FinnTrunk study, for its part, will help to elucidate some aspects of this clinically important question.

### Ethical considerations

The Ethics Committee of the Hospital District of Southwestern Finland has reviewed and accepted the study (ethics committee reference number 93/1801/2020). Based on this approval, institutional permission for the study has been separately applied for and granted by all participating centers.

## Supporting information

**S1 Checklist. SPIRIT 2013 checklist: Recommended items to address in a clinical trial protocol and related documents.**
(PDF)

**S1 File.**
(PDF)

**S2 File.**
(DOCX)

## Author Contributions

**Conceptualization:** Jaakko Viljamaa, Khalil Firoozi, Maarit Venermo, Karoliina Halmesmäki, Harri Hakovirta.

**Funding acquisition:** Harri Hakovirta.

**Methodology:** Jaakko Viljamaa, Maarit Venermo, Matti Pokela, Toni Pihlaja, Karoliina Halmesmäki, Harri Hakovirta.

**Project administration:** Karoliina Halmesmäki, Harri Hakovirta.

**Supervision:** Harri Hakovirta.

**Writing – original draft:** Jaakko Viljamaa, Khalil Firoozi, Maarit Venermo, Matti Pokela, Toni Pihlaja, Karoliina Halmesmäki, Harri Hakovirta.

**Writing – review & editing:** Jaakko Viljamaa, Khalil Firoozi, Maarit Venermo, Matti Pokela, Toni Pihlaja, Karoliina Halmesmäki, Harri Hakovirta.

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
