## [Decision Letter · Decision Letter 0]

16 Mar 2023

PONE-D-23-03544A study protocol for comparing the treatment of varicose tributaries either concomitantly with or separately from endovenous laser ablation of the incompetent saphenous trunk (FinnTrunk Study). A multicenter parallel-group randomized controlled studyPLOS ONE

Dear Dr. Hakovirta,

Thank you for submitting your manuscript to PLOS ONE. After careful consideration, we feel that it has merit but does not fully meet PLOS ONE’s publication criteria as it currently stands. Therefore, we invite you to submit a revised version of the manuscript that addresses the points raised during the review process.

We look forward to receiving your revised manuscript.

Kind regards,

Eyüp Serhat Çalık

Academic Editor

PLOS ONE

3. Please upload a copy of your study protocol that was approved by your ethics committee/IRB as a Supporting Information file. By the study protocol, we mean the complete and detailed plan for the conduct and analysis of the trial approved by the ethics committee/IRB. Please send this in the original language. If this is in a language other than English, please also provide a translation. [https://journals.plos.org/plosone/s/submission-guidelines#loc-guidelines-for-specific-study-types]

Additional Editor Comments:

Dear Authors

I read your work with interest and congratulate you. Overall, it is a well-crafted manuscript, but it needs some improvement. Your manuscript was evaluated by 6 reviewers and below are their recommendations. Please pay attention to the attached documents.  I wish you success.

Reviewers' comments:

Reviewer's Responses to Questions

**Comments to the Author**

1. Does the manuscript provide a valid rationale for the proposed study, with clearly identified and justified research questions?

Reviewer #1: Yes

Reviewer #2: Partly

Reviewer #3: Yes

Reviewer #4: Yes

Reviewer #5: No

Reviewer #6: Yes

2. Is the protocol technically sound and planned in a manner that will lead to a meaningful outcome and allow testing the stated hypotheses?

Reviewer #1: Yes

Reviewer #2: Partly

Reviewer #3: Yes

Reviewer #4: Yes

Reviewer #5: Yes

Reviewer #6: Partly

3. Is the methodology feasible and described in sufficient detail to allow the work to be replicable?

Reviewer #1: Yes

Reviewer #2: No

Reviewer #3: Yes

Reviewer #4: Yes

Reviewer #5: Yes

Reviewer #6: No

4. Have the authors described where all data underlying the findings will be made available when the study is complete?

Reviewer #1: No

Reviewer #2: Yes

Reviewer #3: Yes

Reviewer #4: Yes

Reviewer #5: Yes

Reviewer #6: Yes

5. Is the manuscript presented in an intelligible fashion and written in standard English?

Reviewer #1: Yes

Reviewer #2: Yes

Reviewer #3: Yes

Reviewer #4: Yes

Reviewer #5: Yes

Reviewer #6: No

6. Review Comments to the Author

You may also provide optional suggestions and comments to authors that they might find helpful in planning their study.

Reviewer #1: I have the following questions about this manuscript and protocol:

1. Can the authors explain more about their exclusion criteria for three of their exclusions – Body mass index >35 (this seems too low to me…..I would have expected 40), Prior DVT of the muscular veins of the calf, and Extensive deep vein reflux of the study leg? Please explain these exclusion criteria.

2. Why the exclusions related to the contralateral leg, as described from lines 159-171?

3. The evaluation of the GSV seems very limited to me, with only the thigh GSV evaluated in addition to the SFJ, the AASV and the SSV and SPJ. Typically, the GSV will be evaluated downs its entire length down into the calf. Why do the authors limit the evaluation of the GSV to these locations?

4. In the description of the procedure, the authors do not talk about what they do if they find foam in the deep veins through perforators. Can the authors please discuss this?

5. I am concerned about not having a duplex follow-up within the first week of the procedure. We know that the majority of any thrombotic complications (DVT and EHIT) will occur in the first 7 days postop. By having the first ultrasound at 3 months, you will miss these potential important complications. Can the authors add a one week duplex evaluation to the protocol?

Reviewer #2: The primary outcome measure will be the need for

additional treatment of the varicose tributaries during the follow-up, as judged by the patient is very subjective . I would like to see one or two more objective criteria

Reviewer #3: This is a very interesting study with a very clear and coherent design. The subject matter has an important relevance for daily practice and has not been explored by adequate long-term studies to date.

The text is clearly understandable and well written. The methodology provided for the study is optimal, and the number of cases is sufficiently large. I look forward to the first data.

Reviewer #4: 1. Why 80% power

2. Primary outcome from phlebectomy studies do show difference in re intervention rate.

3. ? Need to include. PREM measure ie satisfaction of care

Reviewer #5: 1. Page 3 line 51-67: as you mentioned in this section, varicose tributaries in VVs patients are very common, even all patients have the tributaries insuffiency, thus nowadays guideline recommend the combined approach for VVs; and the UGFS may accompany with the high recurrence and lower closure rate, thus, the group setting in this study may occur some selective bias. Please revise.

2. Page 2 line 29 and page 5-6 line 105-108 and Table 2: in this study, the author only collect the patients with CEAP classification C2-3, obviously, the selective criteria display some bias, because there are a lot of patients have the C4-5, therefore, this study setting will lose some very important patients, and then the conclusion might be affected. I suggest that the authors make modifications to the selection criteria.

3. Page 7 line 140-144: Many patients with stage C2 have no obvious symptoms, but only show varicose veins. How the author screens out symptomatic patients from a large number of stage C2 patients is a huge workload. Please explain.

4. Page 10 line 229 interventions. In this study the author uses the endovenous laser as the thermal procedure, why don’t you use the radiofrequency ablation; what is more, the author need to clarify whether thermal ablation is only used for the upper knee GSV trunk or the entire GSV, which is very important for the study.

5. In this study, the author did not discuss the setting, advantages and disadvantages of the study, especially with other similar studies. This part needs to be supplemented.

Reviewer #6: Important note: This review pertains only to ‘statistical aspects’ of the study and so ‘clinical aspects’ [like medical importance, relevance of the study, ‘clinical significance and implication(s)’ of the whole study, etc.] are to be evaluated [should be assessed] separately/independently. Further please note that any ‘statistical review’ is generally done under the assumption that (such) study specific methodological [as well as execution] issues are perfectly taken care of by the investigator(s). This review is not an exception to that and so does not cover clinical aspects {however, seldom comments are made only if those issues are intimately / scientifically related & intermingle with ‘statistical aspects’ of the study}. Agreed that ‘statistical methods’ are used as just tools here, however, they are vital part of methodology [and so should be given due importance]. I look at the manuscript in/with statistical view point, other reviewer(s) look(s) at it with different angle so that in totality the review is very comprehensive. However, there should be efforts from authors side to improve (may be by taking clues from reviewer’s comments). Therefore, please do not limit the revision only (with respect) to comments made here.

COMMENTS: I wonder, if the title can be little shorter [say as:”A study protocol for comparing the treatment of varicose disease: A multicenter parallel-group randomized controlled study“ because the aim (as per lines 83-84) is “The aim of the study is to compare two different treatment strategies for varicose disease”]? More specifically, the aim is described in lines 84 onwards anyway.

At few places correction in ‘English’ [sentence construction] is needed. For example: [only 2 are given]

Lines 31-32: Patients will be followed-up at 3 months, 1 year, 3 years, and 5 years.

Possible correction suggested: Patients will be followed-up for 5 years with evaluation recorded at 3 months, 1 year, 3 years, and 5 years [called follow-up visits]. {correct sentence in line 261: The duration of the follow-up will be five years.}

Lines 321-322: Mean values and standard deviations (SD) will be applied for normal distributions.

Possible correction suggested: Mean values and standard deviations (SD) will be estimated (or reported/displayed/tabulated) for variables having normal distributions. {note that we never say: we have apply Mean values and standard deviations (SD)}

Kindly check for the ‘English’ language. Agreed that English is not (may not be) our mother tongue however, remember/mind you that this is a scientific/academic document and so all details should be clearly/correctly communicated (do not take readers’ for granted). You may take help of language professional expert, if needed.

Is the sentence/statement “The first study results will be published in a peer reviewed journal in 2024.” of lines 79-80 necessary? What purpose it serves? Account given (both description & action indicated) in lines 152 to 157 is not understood. This phenomenon (confusing description) is/was, unfortunately, observed/noted at few more places.

Because you stated in lines 323-325 that “the Student’s t-test and ANOVA will be used to compare continuous variables of normal distributions of the values that have previously been tested using the Shapiro–Wilk test.”, please note though the measures/tools used are appropriate [ex. Aberdeen Varicose Vein Questionnaire (AVVQ) score, the Venous Clinical Severity Score 266 (VCSS), health-related quality of life (the EuroQoL EQ-5D-5L) score, pain score based on numeric rating scale (NRS)], most of them are likely to yield data that are in ‘ordinal’ level of measurement [and not in ratio level of measurement for sure {as the score two times higher does not indicate presence of that parameter/phenomenon as double (for example, a Visual Analogue Scales VAS score or say ‘depression’ score)}]. Then application of suitable non-parametric test(s) is/are indicated/advisable [even if distribution may be ‘Gaussian’ (also called ‘normal’)]. Agreed that there is/are no non-parametric test(s)/technique(s) available to be used as alternative in all situation(s) [suitable / most desired/applicable], but should be used whenever/wherever they are available. Therefore, in short use suitable non-parametric test(s)/technique(s) while dealing with data that are in ‘ordinal’ level of measurement even if [despite that] the distribution may be ‘Gaussian’. Testing ‘normality’ in sample [by using any normality test(s)} is not required/desired while dealing with data that are in ‘ordinal’ level of measurement [as most of the normality tests are not valid for ‘ordinal’ data (including Shapiro–Wilk test)].

I suggest authors to check and re-draft the ‘Conclusions’ section (lines 328 to 334). Most of these are would/could not be part of this study. In my opinion, this study has potentials but to rescue this article (which is quite possible), some amount of re-vision (re-drafting) may be needed. However, please do not limit the revision only (with respect) to comments made here. More improvement is expected.

As pointed out in ‘important note’ above “This review pertains only to ‘statistical aspects’ of the study and so ‘clinical aspects’ should be assessed separately/independently [one should carefully consider/look at the clinical implications of the study]. The respected ‘Editor’ may consider accepting only if found ‘clinical implications’ valuable (add to clinical knowledge / positively influence clinical practice). ‘Major revision’ is recommended.

7. PLOS authors have the option to publish the peer review history of their article (what does this mean?). If published, this will include your full peer review and any attached files.

Reviewer #1: No

Reviewer #2: No

Reviewer #3: No

Reviewer #4: No

Reviewer #5: No

Reviewer #6: No

---

## [Author Response · Author response to Decision Letter 0]

15 Apr 2023

Dear editor,

Authors want to thank for the opportunity to revise our present manuscript. The manuscript has been revised according to given critisism and excellent reviewer comments have been answered. Answers are listed bellow. We hope they meet the given comments accurately. We really hope manuscript now meets better the high standard for the articles/protocols published in PLOS one and that manuscript has improved significantly after revision.

Kind regards,

Professor Hakovirta

Reviewers' comments:

Reviewer's Responses to Questions

Comments to the Author

1. Does the manuscript provide a valid rationale for the proposed study, with clearly identified and justified research questions?

Reviewer #1: Yes

Reviewer #2: Partly

Reviewer #3: Yes

Reviewer #4: Yes

Reviewer #5: No

Reviewer #6: Yes

As for providing a rationale for the study, we wished to keep the Introduction and Conclusions [renamed Discussion in this draft] short and concise, in keeping with the nature of the paper. – Our aim was to state only the most relevant facts regarding the evidence underlying, for example, the recommendations of the guidelines referenced in the text [10,11], and to bring up certain aspects of the research question that are relevant for the patient. Obviously, we will return to the previous studies and these questions in more detail when we present our results. 

In our opinion, the research question of the FinnTrunk Study is a valid academic problem. At the very least, it is an important clinical problem that needs to be addressed when making treatment decisions in this patient cohort. As stated in the Introduction, there is a paucity of randomized controlled trials that have looked into this question. There is clearly a need for further studies that examine the timing of tributary treatment as well as the significance of varicose tributaries in the long term and in relation to the recurrence of varicose disease1,2.

1. Aherne TM. et al. Concomitant vs. Staged Treatment of Varicose Tributaries as an Adjunct to Endovenous Ablation: A Systematic Review and Meta-Analysis. Eur J Vasc Endovasc 2020;60:430-442

2. De Maeseneer MG et al. Editor’s Choice – European Society for Vascular Surgery (ESVS) 2022 Clinical Practice Guidelines on the Management of Chronic Venous Disease of the Lower Limbs. Eur J Vasc Endovasc 2022;63:184-267

2. Is the protocol technically sound and planned in a manner that will lead to a meaningful outcome and allow testing the stated hypotheses?

Reviewer #1: Yes

Reviewer #2: Partly

Reviewer #3: Yes

Reviewer #4: Yes

Reviewer #5: Yes

Reviewer #6: Partly

The concerns raised by reviewer #2 regarding the endpoints of the study and by reviewer #6 regarding the statistical methods are addressed, respectively, in our answers to their comments.

3. Is the methodology feasible and described in sufficient detail to allow the work to be replicable?

Reviewer #1: Yes

Reviewer #2: No

Reviewer #3: Yes

Reviewer #4: Yes

Reviewer #5: Yes

Reviewer #6: No

Most of the reviewers agree that the methodology is feasible and described in sufficient detail, the concerns raised by reviewers #2 and #6 are also adressed, in our answers to their comments.

4. Have the authors described where all data underlying the findings will be made available when the study is complete?

The PLOS Data policy<https://journals.plos.org/plosone/s/materials-and-software-sharing> requires authors to make all data underlying the findings described in their manuscript fully available without restriction, with rare exception, at the time of publication. The data should be provided as part of the manuscript or its supporting information, or deposited to a public repository. For example, in addition to summary statistics, the data points behind means, medians and variance measures should be available. If there are restrictions on publicly sharing data—e.g. participant privacy or use of data from a third party—those must be specified.

Reviewer #1: No

Reviewer #2: Yes

Reviewer #3: Yes

Reviewer #4: Yes

Reviewer #5: Yes

Reviewer #6: Yes

As stated in the text (lines 79–80), The first study results will be published in a peer-reviewed journal in 2024. The longer term results (3 and 5 years of follow-up) will be published in due time, but we feel that at the time of publishing the protocol the most relevant publication to be specified in the text is the next one – which, at the same time, is also the one whose publication date can be estimated with some accuracy. Obivously, an estimation of when the longer term results of the study will be expected to be available and published, can be added to the text if need be.

5. Is the manuscript presented in an intelligible fashion and written in standard English?

Reviewer #1: Yes

Reviewer #2: Yes

Reviewer #3: Yes

Reviewer #4: Yes

Reviewer #5: Yes

Reviewer #6: No

The concerns raised by reviewer #6 are addressed in our answers to their comments.

6. Review Comments to the Author

You may also provide optional suggestions and comments to authors that they might find helpful in planning their study.

Reviewer #1: I have the following questions about this manuscript and protocol:

1. Can the authors explain more about their exclusion criteria for three of their exclusions – Body mass index >35 (this seems too low to me…..I would have expected 40), Prior DVT of the muscular veins of the calf, and Extensive deep vein reflux of the study leg? Please explain these exclusion criteria.

Obesity is arguably a risk factor for both venous disease and the skin changes associated with it3–6, and it has also been suggested that class III obesity (BMI >40) could contribute to these skin changes even in the absence of venous disease7. In addition to this, the outcomes seem to progressively worsen with a BMI > 35 for patients undergoing treatment for venous disease8. Taking all these aspects and the long follow-up time of our study into consideration, it was felt that the exclusion criterion BMI >35 could decrease the effect of obesity as a confounding factor. At the same time, BMI >35 is also the cut-off point for class II obesity.

3. Danielsson G et al. The influence of obesity on chronic venous disease. Vasc Endovascular Surg 2002;36:271-276

4. Robertson L et al. Risk factors for chronic ulceration in patients with varicose veins: a case control study. J Vasc Surg 2009;49:1490-1498

5. Davies HO et al. Obesity and lower limb venous disease - The epidemic of phlebesity. Phlebology 2017;32:227-233

6. Salim S et al. Global Epidemiology of Chronic Venous Disease. Ann Surg. 2021;274: 971-976

7. Padberg F Jr et al. Does severe venous insufficiency have a different etiology in the morbidly obese? Is it venous? J Vasc Surg 2003;37:79-85

8. Deol ZK et all. Effect of obesity on chronic venous insufficiency treatment outcomes. J Vasc Surg Venous Lymphat Disord 2020;8:617-628

Although in the clinical setting patients with a prior deep vein thrombosis are routinely treated with endovenous techniques (combined with thromboprofylaxis post-procedure), it was decided to exclude these patients from the study for safety reasons. As this exclusion criterion will be executed based mainly on patient interview it was felt that the decision would be more straightforward and unequivocal to make and less prone to recall bias if all lower extremity thrombotic events (bar superficial vein thrombosis) were combined to the criterion. – Superficial vein thrombosis can usually be easily differentiated from deep vein thrombosis based on its clinical characteristics and treatment, and in Finland, thrombosis of the muscular veins of the calf is routinely treated with the same regimen as deep vein thrombosis. 

Extensive deep vein reflux, i.e., reflux of both the CFV and PV, is an exclusion criteria because it was reasoned that it could be a cause for residual symptoms even after the superficial reflux had been treated and therefore cause bias in the quality of life questionnaires. It should be noted, though, that because prior deep vein thrombosis is an exclusion criterion, as well, the number of patients excluded because of extensive deep vein reflux should be low.

2. Why the exclusions related to the contralateral leg, as described from lines 159-171?

It was reasoned that if there were significant symptoms due to venous disease in the contralateral leg, it could cause bias in the quality of life questionnaires. 

The evaluation of the GSV seems very limited to me, with only the thigh GSV evaluated in addition to the SFJ, the AASV and the SSV and SPJ. Typically, the GSV will be evaluated downs its entire length down into the calf. Why do the authors limit the evaluation of the GSV to these locations?

The reflux of the below-knee GSV will be recorded under any other veins as indicated (lines 190-191 in Revised Manuscript with Track Changes). True, the diameter of the distal GSV will not be recorded, but it was felt that the diameter of the thigh GSV was the more important variable – all the way from the decision to use thermal ablation for the treatment. Secondly, the diameter of the below-knee GSV is not routinely measured in the clinical setting, and it was our aim to keep the DUS protocol as close to the clinical setting as possible. 

3. In the description of the procedure, the authors do not talk about what they do if they find foam in the deep veins through perforators. Can the authors please discuss this?

It is standard procedure in the participating centres that the patency of the deep veins is verified only in the segment adjacent to the junction of the treated superficial vein with the deep vein. The deep veins of the calf are not routinely examined with DUS at the end of the procedure.

4. I am concerned about not having a duplex follow-up within the first week of the procedure. We know that the majority of any thrombotic complications (DVT and EHIT) will occur in the first 7 days postop. By having the first ultrasound at 3 months, you will miss these potential important complications. Can the authors add a one week duplex evaluation to the protocol?

We are aware that most guidelines recommend a DUS follow-up during the first week after the procedure, for the very reasons the reviewer has mentioned. However, in the clinical setting this is not routinely undertaken in any of the participating centres and it would have been difficult to justify in a clinical study. That said, the participants are instructed to contact the treating unit with low threshold should they experience any symptoms consistent with deep vein thrombosis. As for adding an early DUS follow-up to the protocol, the study is already under way (lines 78–79). 

Reviewer #2: The primary outcome measure will be the need for additional treatment of the varicose tributaries during the follow-up, as judged by the patient is very subjective . I would like to see one or two more objective criteria

 The treatment indications of C2–3 varicose disease, i.e., venous disease that has not caused any skin changes, are relative and based on the symptoms of the patient. Most likely, though usually left unspoken, also the cosmetic result is of importance for the patient. Accordingly, the primary outcome measure is intentionally subjective: it is supposed to reflect those aspects of the treatment result that are important from the patient’s perspective. In addition to the primary outcome measure, there are two objective secondary outcome measures and several objective outcome measures of interest in the study. 

Reviewer #3: This is a very interesting study with a very clear and coherent design. The subject matter has an important relevance for daily practice and has not been explored by adequate long-term studies to date. The text is clearly understandable and well written. The methodology provided for the study is optimal, and the number of cases is sufficiently large. I look forward to the first data.

–

Reviewer #4: 1. Why 80% power

2. Primary outcome from phlebectomy studies do show difference in re intervention rate.

3. ? Need to include. PREM measure ie satisfaction of care

1. 

Generally, both 80% and 90% power are used. As stated in the text, the sample size was intentionally increased to meet possible dropouts, and, actually, the sample size is also valid for 90% power, as calculated according to the results of the meta-analysis by Aherne et al [13]. 

2.

That is correct, though interestingly, the meta-analysis by Aherne et al. [13] found no significant difference between the combined and staged approaches with regard to re-intervention rate in their sensitivity analysis limited to the three available RCTs [14-16]. However, we wished to primarily draw attention to the two aspects of this research question that have relevance regarding the rationale of our study: that there is a paucity of RCTs to backup the recommendations of the guidelines [10,11] for the combined approach (lines 53-54 in the Introduction) and that both approaches (combined and staged) can be considered acceptable (lines 51-53 in the Introduction) – and that most likely, they are both used in the clinical setting, despite the guidelines (lines 340-345 in the Conclusions [renamed Discussion in this draft] in Revised Manuscript with Track Changes). 

3. 

Data pertaining to the patient’s experience of the treatment will be collected at 1 year with two questions (“Was the treatment you were given of good quality?” and “Did the treatment you were given meet your expectations?”) (lines 277-279 in Revised Manuscript with Track Changes). Though these questions are defined as PROMs in the protocol, it could argued that they also cover PREMs, at least partly. As for adding more questions, the study is already under way (lines 78–79).

Reviewer #5: 1. Page 3 line 51-67: as you mentioned in this section, varicose tributaries in VVs patients are very common, even all patients have the tributaries insuffiency, thus nowadays guideline recommend the combined approach for VVs; and the UGFS may accompany with the high recurrence and lower closure rate, thus, the group setting in this study may occur some selective bias. Please revise.

If this comment relates to not treating the varicose tributaries in group one, that is one of the aims of the study – to find out if varicose tributaries left untreated will expose the patient to the risk of recurrence of the varicose disease. As mentioned in our answer to the previous question, the study is already under way. That being the case, we are unfortunately unable to make any changes regarding the treatment.

Page 2 line 29 and page 5-6 line 105-108 and Table 2: in this study, the author only collect the patients with CEAP classification C2-3, obviously, the selective criteria display some bias, because there are a lot of patients have the C4-5, therefore, this study setting will lose some very important patients, and then the conclusion might be affected. I suggest that the authors make modifications to the selection criteria.

As stated previously, the study is already under way, and in the last meeting of the study group it was decided that the recruitment was progressing according to plans and that there would be no need to recruit patients in the CEAP clinical classes C4a–4b (lines 115-116).

The decision to limit the recruitment to patients in the CEAP clinical classes C2–3 was done in order to have a cohort where treatment decision would be based purely on the symptoms (as opposed to skin changes in the CEAP clinical classes ≥C4). It is acknowledged that this will limit the extrapolation of our results to patients in higher CEAP clinical classes – if not necessarily cause bias – and we will most definitely address this limitation when the study results are published. However, it can also be questioned whether staged approach is acceptable when treating patients who have skin changes caused by venous disease (as opposed to treating patients without skin changes) or argued that when a patient has a healed venous ulcer the treatment goal should be to abolish all clinically relevant reflux routes and to use combined approach to this end. We will get back to these questions as well when we publish the study results.

2. Page 7 line 140-144: Many patients with stage C2 have no obvious symptoms, but only show varicose veins. How the author screens out symptomatic patients from a large number of stage C2 patients is a huge workload. Please explain.

The study will screen and recruit patients referred to the participating centres (lines 106-107) by GPs, occupational health physicians, and private practitioners who are aware of the general indications for treatment of the varicose disease in the public sector in Finland – that is, patients will not be treated for purely cosmetic reasons. In other words, patients who have varicose veins but who do not present any obvious venous symptoms will not be – at least, for the most part – referred for consideration of treatment. 

Page 10 line 229 interventions. In this study the author uses the endovenous laser as the thermal procedure, why don’t you use the radiofrequency ablation; what is more, the author need to clarify whether thermal ablation is only used for the upper knee GSV trunk or the entire GSV, which is very important for the study.

Endovenous laser ablation (starting on line 238 in Revised Manuscript with Track Changes) was chosen because it is the primary method of endovenous thermal ablation in use in the participating vascular surgery units (or, the only method in certain units). We are not sure if this needs to be addressed in the protocol, as we are confident readers will make the assumption that endovascular laser ablation was chosen simply for such pragmatic reasons. For comparison, endovenous laser ablation was used in two of the RCTs [14,15] referenced in the text and radiofrequency ablation in the third [16], but none of those three papers provided any rationale for their choice of method. On a sidenote, adhering to one method of thermal ablation should of course have the added advantage of decreasing variation related to the procedure.

As for the treated segment of GSV, please refer to lines 244-246 in Revised Manuscript with Track Changes.

3. In this study, the author did not discuss the setting, advantages and disadvantages of the study, especially with other similar studies. This part needs to be supplemented.

As stated previously, we wished to keep the Introduction and Conclusions [renamed Discussion in this draft] short and concise. The setting is defined in the text (lines 71-75, 106-107). True, the advantages and disadvantages are not discussed, but the reason for that is what the reviewer has pointed out – this should be undertaken especially in relation to previous similar studies. At this point, that would require at least a certain amount of speculation and also involve the risk of making asumptions that might later prove wrong. For these reasons we feel that these comparisons are better made when have our results in, and we will be making these comparisons in our forthcoming article(s), of course. By not making them now, we will also avoid repeating / copying comparisons and interpretations we have already published, but which will definitely have to be discussed – by convention – when the results of a study are published.

Reviewer #6: Important note: This review pertains only to ‘statistical aspects’ of the study and so ‘clinical aspects’ [like medical importance, relevance of the study, ‘clinical significance and implication(s)’ of the whole study, etc.] are to be evaluated [should be assessed] separately/independently. Further please note that any ‘statistical review’ is generally done under the assumption that (such) study specific methodological [as well as execution] issues are perfectly taken care of by the investigator(s). This review is not an exception to that and so does not cover clinical aspects {however, seldom comments are made only if those issues are intimately / scientifically related & intermingle with ‘statistical aspects’ of the study}. Agreed that ‘statistical methods’ are used as just tools here, however, they are vital part of methodology [and so should be given due importance]. I look at the manuscript in/with statistical view point, other reviewer(s) look(s) at it with different angle so that in totality the review is very comprehensive. However, there should be efforts from authors side to improve (may be by taking clues from reviewer’s comments). Therefore, please do not limit the revision only (with respect) to comments made here.

COMMENTS: I wonder, if the title can be little shorter [say as:”A study protocol for comparing the treatment of varicose disease: A multicenter parallel-group randomized controlled study“ because the aim (as per lines 83-84) is “The aim of the study is to compare two different treatment strategies for varicose disease”]? More specifically, the aim is described in lines 84 onwards anyway.

According to the submission guidelines given by PLOS ONE, the full title should (among other things) be ”specific and descriptive”, but it is true that the title could perhaps be shorter. However, many readers scan only the titles, and we would rather leave the title as it is, because it now describes our study in two sentences, without the need for the reader to read any further. The title can be changed, of course, if the editor wishes so.

At few places correction in ‘English’ [sentence construction] is needed. For example: [only 2 are given] Lines 31-32: Patients will be followed-up at 3 months, 1 year, 3 years, and 5 years.

Possible correction suggested: Patients will be followed-up for 5 years with evaluation recorded at 3 months, 1 year, 3 years, and 5 years [called follow-up visits]. {correct sentence in line 261: The duration of the follow-up will be five years.}

Lines 321-322: Mean values and standard deviations (SD) will be applied for normal distributions. Possible correction suggested: Mean values and standard deviations (SD) will be estimated (or 

reported/displayed/tabulated) for variables having normal distributions. {note that we never say: we have apply Mean values and standard deviations (SD)}

Kindly check for the ‘English’ language. Agreed that English is not (may not be) our mother tongue however, remember/mind you that this is a scientific/academic document and so all details should be clearly/correctly communicated (do not take readers’ for granted). You may take help of language professional expert, if needed.

As is obvious, English is not our mother tongue. Because of this, the manuscript was language checked by a Finnish speaking translator in the early stages, and before submission the final draft was checked by a Scottish professional science editor who according to his bio has revised over 1500 research manuscripts. However, we would be more than happy to go through the manuscript with him line by line and word by word at this stage. For that purpose we would kindly ask you to highlight all the passages that should, in your opinion, be revised. Absolutely no explanations needed. 

Is the sentence/statement “The first study results will be published in a peer reviewed journal in 2024.” of lines 79-80 necessary? What purpose it serves? Account given (both description & action indicated) in lines 152 to 157 is not understood. This phenomenon (confusing description) is/was, unfortunately, observed/noted at few more places.

The sentence on lines 79-80 is necessary. According to the submission guidelines given by PLOS ONE, the study protocol should include information on ”where and when the data will be made available”. Please also refer to Question 4. in the beginning of this document.

As stated in the inclusion criteria, the study will recruit patients with unilateral symptomatic varicose disease. That said, it is extremely common for the patient to present with varicose veins also in the contralateral leg, and to present with varicose disease indicating treatment – because of symptoms and/or skin changes – in the contralateral leg is far from rare. The lines 152-157 described the criteria by which patients with bilateral disease may be recruited to the study. Please refer to the revised text (lines 151-163 in Revised Manuscript with Track Changes). Would those revisions help?

If there are more passages in the text that would benefit from a second look and possibly a revision, we would appreciate if they were pointed out to us – we have, of course, written the text from a vascular surgeon’s perspective, and things that are clear for us by default might not be that for the general readership.

Because you stated in lines 323-325 that “the Student’s t-test and ANOVA will be used to compare continuous variables of normal distributions of the values that have previously been tested using the Shapiro–Wilk test.”, please note though the measures/tools used are appropriate [ex. Aberdeen Varicose Vein Questionnaire (AVVQ) score, the Venous Clinical Severity Score 266 (VCSS), health-related quality of life (the EuroQoL EQ-5D-5L) score, pain score based on numeric rating scale (NRS)], most of them are likely to yield data that are in ‘ordinal’ level of measurement [and not in ratio level of measurement for sure {as the score two times higher does not indicate presence of that parameter/phenomenon as double (for example, a Visual Analogue Scales VAS score or say ‘depression’ score)}]. Then application of suitable non-parametric test(s) is/are indicated/advisable [even if distribution may be ‘Gaussian’ (also called ‘normal’)]. Agreed that there is/are no non-parametric test(s)/technique(s) available to be used as alternative in all situation(s) [suitable / most desired/applicable], but should be used whenever/wherever they are available. Therefore, in short use suitable non-parametric test(s)/technique(s) while dealing with data that are in ‘ordinal’ level of measurement even if [despite that] the distribution may be ‘Gaussian’. Testing ‘normality’ in sample [by using any normality test(s)} is not required/desired while dealing with data that are in ‘ordinal’ level of measurement [as most of the normality tests are not valid for ‘ordinal’ data (including Shapiro–Wilk test)].

Thank you for pointing out a clear mistake. Please refer to the revised text (lines 330-337 in Revised Manuscript with Track Changes).

I suggest authors to check and re-draft the ‘Conclusions’ section (lines 328 to 334). Most of these are would/could not be part of this study. In my opinion, this study has potentials but to rescue this article (which is quite possible), some amount of re-vision (re-drafting) may be needed. However, please do not limit the revision only (with respect) to comments made here. More improvement is expected.

”Most of these [conclusions] are would/could not be part of this study.” That is absolutely correct. The heading was changed from Conclusions to Discussion (line 340 in Revised Manuscript with Track Changes). 

As pointed out in ‘important note’ above “This review pertains only to ‘statistical aspects’ of the study and so ‘clinical aspects’ should be assessed separately/independently [one should carefully consider/look at the clinical implications of the study]. The respected ‘Editor’ may consider accepting only if found ‘clinical implications’ valuable (add to clinical knowledge / positively influence clinical practice). ‘Major revision’ is recommended.

---

## [Editor Report · Decision Letter 1]

3 May 2023

A study protocol for comparing the treatment of varicose tributaries either concomitantly with or separately from endovenous laser ablation of the incompetent saphenous trunk (FinnTrunk Study). A multicenter parallel-group randomized controlled study

PONE-D-23-03544R1

Dear Dr. Hakovirta,

We’re pleased to inform you that your manuscript has been judged scientifically suitable for publication and will be formally accepted for publication once it meets all outstanding technical requirements.

Kind regards,

Eyüp Serhat Çalık

Academic Editor

PLOS ONE
---

## [Editor Report · Acceptance letter]

12 May 2023

PONE-D-23-03544R1 

A study protocol for comparing the treatment of varicose tributaries either concomitantly with or separately from endovenous laser ablation of the incompetent saphenous trunk (FinnTrunk Study). A multicenter parallel-group randomized controlled study 

Dear Dr. Hakovirta:

I'm pleased to inform you that your manuscript has been deemed suitable for publication in PLOS ONE. Congratulations! Your manuscript is now with our production department. 

Kind regards, 

on behalf of

Dr. Eyüp Serhat Çalık 

Academic Editor

PLOS ONE